# Room Temperature Polymorphism in WO_3_ Produced by Resistive Heating of W Wires

**DOI:** 10.3390/nano13050884

**Published:** 2023-02-26

**Authors:** Beatriz Rodríguez, Jaime Dolado, Jesus López-Sánchez, Pedro Hidalgo, Bianchi Méndez

**Affiliations:** 1Departament Physics of Materials, Faculty of Physical Sciences, University Complutense of Madrid, 28040 Madrid, Spain; 2European Synchrotron Radiation Facility, 38043 Grenoble, France; 3Spanish CRG BM25 Beamline-SpLine at the European Synchrotron Radiation Facility (ESRF), 38043 Grenoble, France

**Keywords:** tungsten trioxide, polymorphism, Joule heating, electromigration

## Abstract

Polymorphous WO_3_ micro- and nanostructures have been synthesized by the controlled Joule heating of tungsten wires under ambient conditions in a few seconds. The growth on the wire surface is assisted by the electromigration process and it is further enhanced by the application of an external electric field through a pair of biased parallel copper plates. In this case, a high amount of WO_3_ material is also deposited on the copper electrodes, consisting of a few cm${}^{2}$ area. The temperature measurements of the W wire agrees with the values calculated by a finite element model, which has allowed us to establish the threshold density current to trigger the WO_3_ growth. The structural characterization of the produced microstructures accounts for the γ-WO_3_ (monoclinic I), which is the common stable phase at room temperature, along with low temperature phases, known as δ-WO_3_ (triclinic) on structures formed on the wire surface and ϵ-WO_3_ (monoclinic II) on material deposited on external electrodes. These phases allow for a high oxygen vacancies concentration, which is interesting in photocatalysis and sensing applications. The results could help to design experiments to produce oxide nanomaterials from other metal wires by this resistive heating method with scaling-up potential.

## 1. Introduction

Tungsten trioxide, WO_3_, is an attractive semiconducting material widely studied for applications in photochromic devices, photocatalysis and chemical sensing, among others [1,2]. Its ideal crystalline structure can be depicted as a perovskite-like ReO_3_ structure, with WO_6_ octahedra as building blocks [2]. However, depending on the temperature, these octahedra may suffer slight distortions, leading to a variety of crystal phases due to the strong interplay between the vibrational and electronic structure. In particular, five crystalline phases have been reported, which are the monoclinic I (γ-WO_3_), in a the stable phase at room temperature. The low temperature phases, monoclinic II (ϵ-WO_3_, from 5 to 230 K) and triclinic (δ-WO_3_, from 240–290 K) [3,4,5], are less known than the orthorhombic (β-WO_3_, from 290 to 600 K) and tetragonal phases (α-WO_3_, from 600 to 1100 K), which are high temperature phases [5]. Since the physical properties strongly depend on the crystalline structure, the variety of WO_3_ phases widen the applications field of this oxide. As an example, ϵ-WO_3_ exhibits a ferroelectric behaviour due to the slight displacement of the central W ion in the WO_6_ octahedra, which would be of interest in memristor applications or sensing applications [6]. Up to now, the phase transitions sequence from one phase to another, in particular for low temperature phases, has been probed to be quite sensitive to the particular synthesis route and the crystal size [4,7]. For instance, it has been reported that phase transitions of WO_3_ in the low temperature regime occur at different temperatures in sintered material compared to gas-evaporated microcrystals, leading to an stabilization of the triclinic phase at room temperature [4,5].

Concerning the synthesis of metal oxides’ micro- and nanostructures, the Joule heating of metallic wires exposed to air/oxygen has been proposed as an alternative method to those based on sintering or thermal evaporation processes. The resistive heating of copper, iron, vanadium or zinc wires up to temperatures in the 720–800 K range, depending on the metal, has led to the formation of CuO, Fe_2_O_3_, V_2_O_5_ or ZnO nanowires [8,9,10,11], or quasi two-dimensional MoO_3_ flakes [12,13]. Regarding WO_3_, the Joule heating of a tungsten filament has been used as a way to provide W species for the growth of the hot filament chemical vapor deposition (HFCVD) of WO_3_ thin films deposited on a substrate [14]. Alternatively, tungsten oxide nanowires have been fabricated by the Joule heating of W thin films with a designed micropattern that allows the formation of nanowires at the narrower sections of the design [15]. In most of the reported cases, as in these examples, the Joule heating serves are one step in the synthesis route that usually requires vacuum chambers or foreign substrates. Herein, we will take advantage of the Joule heating of a W wire to produce a high amount of WO_3_ microstructures directly on the W filament surface or, eventually, on an external metal plate by means of a setup that does not need vacuum conditions, gas flow, or the additional heating of substrates. During the electrical current passing through the wire, some lattice expansion due to both the thermal and electron flow occurs, inducing the ions’ movement, which is known as the electromigration phenomenon [16,17]. This process has been traditionally considered a risk failure in the electrical transport through metallic wires, since it promotes the formation of hillocks and voids in the material, which are clearly detrimental for the electrical conduction. However, the accumulation of atoms at some points in the wire and the high temperatures achievable during the current flow may favor the conditions for the synthesis of metal oxide nanomaterials on the surface wire before it breaks down.

In this work, we use the Joule heating of metallic tungsten wires exposed to air ambient conditions to produce WO_3_ micro- and nanocrystals in a fast and low-cost way, avoiding the use of furnaces, catalysts and vacuum chambers, which are usually needed in other synthesis methods. The high melting point of tungsten (3143 K) in comparison with the melting point of the above mentioned metals (all of them were below 3000 K) would make the achievement of WO_3_ nanostructures a challenging task by this method. To achieve this goal, the resistive heating of the W wire as a function of the electrical current has been simulated by COMSOL Multiphysics, which take into account the heat and charge transfer within the wire. The temperature-calculated values agreed with the experimental measurements. The density currents required are in the range of j∼10${}^{4}$–10${}^{5}$ A/cm${}^{2}$. In addition, the resistive heating experiments have also been carried out under an external electrical field applied perpendicular to the wire, which in the case of MoO_3_ enhanced the growth process [13]. The characterization of the synthesised WO_3_ material by means of several beam injection techniques has revealed the presence of several crystalline phases with good crystal quality, as demonstrated in the analysis of the experimental data. Our results demonstrate that besides the formation of a high density of WO_3_ micro- and nanostructures, the method provides an effective and rapid way to achieve low temperature structural phases of WO_3_. This could be of high interest in the applications of this oxide. Furthermore, the knowledge gained from the simulation of the process allows us to anticipate the application of this method to other metal wires in order to produce the corresponding metal oxide nanomaterials with a scaling-up potential in a fast and cost-effective way.

## 2. Materials and Methods

The WO_3_ nanostructures were produced by the resistivity heating of W wires of a 0.5 mm diameter (Sigma Aldrich, 99.9% purity) under atmospheric conditions. A Keithley 2400 Series SourceMeter capable of supplying electrical currents up to 20 A was employed as the current source. The local temperature of the wire was assessed by an Infratherm pyrometer calibrated with the emissivity parameters of tungsten. The calculation of the temperature profile along the wire has been conducted with the aid of COMSOL software (electrical current, solid heat transfer and electromagnetic heating packages). The experiment was also conducted under an external electric field applied perpendicularly to the wire through two copper plates that were separated by 6 mm.

The structural features of the produced samples were assessed by XRD diffraction (X’Pert Pro Alpha 1 instrument in the CAI Diffraction facility at the UCM, using an X-ray of the Cu line $K_{\alpha }1$) and Raman spectroscopy (HoribaJobinYbon LabRam), resulting in the identification of several crystalline phases using a 325 nm laser.

Furthermore, scanning electron microscopy (SEM) and transmission electron microscopy (TEM) were used to study the morphology and structural phases of individual microstructures. To that purpose, a FEI Inspec or a Prisma Thermofisher SEM instruments were employed. The TEM microscopy was carried out in a JEOL 300 at ICTS-CNM facility in the UCM. The X-ray photoemission measurements were performed in the ESCAmicroscopy beamline, with a beam energy of 620 eV, at the Elettra synchrotron. The energy calibration was made with the aid of a gold foil. A complementary crystal structure was also studied by high-resolution XRD performed at the CRG BM25 SpLine beamline at the European Synchrotron Radiation Facility (ESRF) in Grenoble (France). The incident X-ray radiation used was 22.3 keV. The powders were introduced in a spinning quartz capillary (0.5 mm diameter) and the acquisition was performed in the 2θ range 5–30º, with a step size of 0.0075º. A 2D photon-counting X-ray MAXIPIX detector was used and the data were processed with the BINoculars software [18,19].

## 3. Results and Discussion

### 3.1. Synthesis Method

It is well-known that the flow of electrical current through a conductor induces Joule heating. This heating will depend on the equilibrium and transport properties of the material, such as the density (ρ), specific heat ($c_{p}$), the electrical (σ) and the thermal conductivity ($\kappa_{T}$). The electron flow at high current densities could lead to cations’ transport in the conductor via the momentum transfer, which is known as the electromigration process. Herein, we aim at harvesting this phenomenon to produce WO_3_ micro- and nanostructures on the surface of W wires under ambient conditions when an electrical current is going through the wire.

Figure 1a shows the layout of the designed setup for the resistive heating experiments along with a photograph of the actual system. A 6 cm W wire is placed between two electrical contacts to where the current source is applied. The conductor elements stand over a PVC insulator base. Measurements are conducted under air conditions and at room temperature. The temperature of the whole system when the electrical current is passing through the wire was calculated taking into account both electrical ($\vec{J}=\sigma \vec{E}$) and thermal transport ($\vec{Q}=\kappa_{T}\nabla T$) across a specific conductor, and is governed by the following equations:(1)$$\begin{array}{c} \nabla \vec{Q}=\vec{J}\cdot \vec{E}\\ Q=\rho c_{p}\frac{\partial T}{\partial t}-\nabla \left(\kappa_{T}\nabla T\right) \end{array}$$
where $\vec{J}$ is the total electrical density current, $\vec{E}$ is the electric field and *Q* is the heat energy. The temperature reached at the stationary state can be obtained by solving the coupled equations system in Equation (Equation 1) with a finite element model. In this case, the material properties (density, specific heat, the electrical and thermal conductivity, and dielectric constant) of tungsten as a function of the temperature were taken from the literature [20,21,22].

In electronic conduction, the heat generated by the current flow is the Joule heating, and electromigration is the atomic diffusion driven by a high current density in the direction of the electron flow. Joule heating and electromigration are two coupled mechanisms. On the one side, Joule heating will increase the temperature of the sample that, in the case of filaments or wires, will create a temperature gradient from the central part towards the edges because of heat dissipation in the material. This process affects the atomic diffusion on the material. On the other hand, at high temperatures, a faster atomic diffusion will drive a faster electromigration’s rate in turn [17,23]. The electromigration process requires a high current density. Actually, current densities in the range of 10${}^{4}$ A/cm${}^{2}$ were reported as threshold values to achieve enough electron flow-induced strain to cause the electromigration process in the Cu stripes [16].

Our objective is to achieve an accumulation of cations at the wire surface that could act as nucleation sites for the growth of WO_3_. The simulation results of the electron and heat transport along the tungsten wire have allowed us to target the current values required to achieve the high temperatures enough for the tungsten oxide crystallization. After several experiments, we conclude that there is a critical current density below which no growth will occur; for our treatments, the minimum current that needed to be applied is 16.5 A. This means a density current of 7.5 ×$10^{4}$ A/cm${}^{2}$. If we use a higher current, the time needed for the growth will be reduced. The higher temperatures are expected at the central part of the wire, since more heat dissipation occurs closer to the metal contacts ends. Figure 1b displays the local temperature of the whole system calculated when a current of 16.5 A is applied after 10 s. The scale bar shows a local temperature of 1480 K at the centre of the wire. We performed simulations at several currents to assess the local heating response of the W wire. Figure 1c shows the temperature profiles along the wire when density currents vary between 5–8 ×$10^{4}$ A/cm${}^{2}$. It is observed that the central region of the wire, between 2 and 4 cm depending on the current, is well above 1000 K. Moreover, the native oxide layer at the surface of the W wire may prevent further atomic diffusion, which leads to an accumulation of atoms on the wire surface that favor the nucleation of metal oxide microstructures.

Figure 2a shows the secondary electron (SE) image of the wire after 50 s sustaining a current of 16.5 A. The time for the synthesis could be extended, but at a certain point the wire is going to break because of the material loss by the electromigration process dragging. Under higher currents than 16.5 A, the time needed for the growth is reduced to few seconds. Finally, above 18 A, the wire breaks down almost immediately. The temperature measured with the Infrared thermometer is 1510 K, which fairly agrees with the calculated one (1480 K). The Joule heating of the wire has caused the formation of well-crystallized structures over about 4 cm in the central part of the wire. According to the simulations (Figure 1c), the temperature in this region is between 1000–1510 K for 16.5 A, which propitiates the nucleation and formation of the oxide crystalline structures. In spite of the thermal gradient along the wire, the morphology of the structures formed on the wire are rather uniform, as is shown in Figure 2a, although a lower density of WO_3_ structures is found as we go further from the central part of the wire. The detailed morphology of these micro- and nanostructures are shown in Figure 2b,c. The most common shape corresponds to hierarchical microstructures that consist of several tapered rods, often emerging from the same point and built by stacked plates of decreasing size, as is shown in Figure 2b,c. Eventually, small needles are also formed on the lateral plates as secondary growth (Figure 2d). These oxide structures form a shell around the metallic wire of a thickness of about a hundred micron, being easy to separate from the wire. Moreover, the remaining W wire under the oxide layer can be reused in subsequent new growths.

In addition, experiments were also conducted under an external electric field applied perpendicularly to the W wire by placing two copper electrodes, as shown in Figure 1a. This approach not only adds a drift component to the atomic flux towards the wire surface, but also a cations flight from the surface to the electrode plates. This process adds another source for the nucleation and growth of WO_3_ nanostructures. As a result, besides the microstructures formed on the wire surface, a big amount of micro- and nanostructures are also deposited on the external copper plates, giving rise to a thin layer of structures of a few hundred microns on the electrode. Figure 2e shows the secondary electron image of these WO_3_ nanostructures. It is worth noticing that new morphologies, as microspheres, appear on the electrodes in comparison with structures formed on the surface wire. These first results demonstrate that the Joule heating of tungsten wires under suitable electrical density currents bring about WO_3_ micro- and nanostructures of several shapes, which would imply, as well, several crystalline phases.

### 3.2. Structural Characterization

In order to assess the crystalline structure of the achieved WO_3_ microstructures, X-ray diffraction experiments were carried out. To this purpose, the structures were transferred to a silicon substrate. Figure 3a shows the XRD diffractogram recorded under the gracing incident for three representative samples: WO_3_ formed on the wire surface without an external electric field, with an external electric field and WO_3_ deposited on copper electrodes, revealing the presence of multiple crystalline phases. Some peaks are common to the three samples due to the small variations among the phases [4]; thus, due to the coexistence of phases and the overlapping of the patterns, the analysis becomes difficult. However, focusing on the 22–25º interval, the characteristic peaks of each phase can be resolved. Figure 3b shows the zoom up of XRD data in this angle interval for the three samples in which peaks assigned to the ϵ-WO_3_ (ICSD 10872385), γ-WO_3_ (ICSD 80056) and δ-WO_3_ (ICSD 1620) phases are marked with stars, squares and diamonds, respectively. The XRD results show that WO_3_ formed on the wire consists of a mixture of γ-WO_3_ and δ-WO_3_ phases, while WO_3_ deposited on copper electrodes contains the ϵ-WO_3_ phase, as well.

To corroborate these results, synchrotron XRD measurements were also carried out at the ESRF (BM25-SpLine beamline) in WO_3_ structures deposited on copper electrodes and on the wire surface. To that end, samples were inserted in a quartz capillary that rotates during the measurements, to avoid any anisotropic effects. Synchrotron radiation XRD results of two representative samples: structures formed on the W wire surface under an external electric field and on the external copper electrodes have been performed. The results are shown in Figure 3c, in which the yellow plot stands for WO_3_ from the the wire surface and the pink plot for structures deposited on electrodes. From these results, there is also a clear evidence of the presence of the ϵ-phase in the WO_3_ microstructures deposited on copper electrodes.

The polymorphous nature of the WO_3_ microstructures obtained by resistive heating was also assessed from micro-Raman measurements performed in a confocal microscope. Figure 4 shows the Raman spectra of structures from the electrodes (green and light blue lines) and from the wire surface (purple and dark blue lines). The main characteristic Raman bands of the γ-WO_3_, at 133, 272, 716 and 803 cm${}^{-1}$, are present in all the analyzed samples. The low energy bands, 133 cm${}^{-1}$ and 272 cm${}^{-1}$, are associated to O-W-O lattice vibrations and bending modes, respectively, while the high energy bands (716–803 cm${}^{-1}$) may be related to the O-W-O stretching vibrational modes. This phase has been reported to be stable at room temperature in microcrystalline WO_3_ [7]. Besides this phase, a careful inspection of the high energy region highlighted with a red square in Figure 4 reveals that characteristic modes of the reported low temperature phases can also be resolved [4,5,7]. In particular, a band at 605 cm${}^{-1}$, marked with a diamond and characteristic of the δ-WO_3_, is observed in some structures formed on the surface wire, and bands at 636 and 673 cm${}^{-1}$, marked with stars and assigned to the ϵ-WO_3_, are detected in structures deposited on the copper electrodes. These results agree with the above-shown diffraction data.

In order to complete the structural characterization, a TEM study of individual microstructures is conducted. Figure 5a shows a low magnification TEM image of the nanostructures deposited on the electrode, in which two types of morphologies, rounded and octahedron-shaped particles marked with circles, are clearly identified. Figure 5b,c show the HRTEM image and the SAED pattern from an spherical particle. Diffraction spots have been identified under the [001] zone axis of the ϵ-phase of WO_3_. Figure 5d shows the sketch of the atomic structure in this orientation and agrees with experimental results. An analogue HRTEM and SAED analysis has been made for the octahedral particles, resulting in the δ-phase as the WO_3_ structural order (Figure 5e–g).

The above characterization results allow us to draw the following conclusions. The resistive heating of W wires under suitable density currents (5–8 × 10${}^{4}$ A/cm${}^{2}$) and assisted by an external electric field, gives rise to a high amount of WO_3_ microstructures adopting several shapes. This growth process is far from the equilibrium and yields polymorphic material with up to three structural phases, that otherwise would require low temperature conditions. The γ-WO_3_ phase, which is stable at room temperature, is the dominant one in all samples, as could be expected. However, the δ-WO_3_ phase is also found both in structures formed on the wire surface and on the electrodes, depending on the local area investigated. In addition, the ϵ-WO_3_ phase, which has only been reported at low temperatures, has also been identified in microstructures deposited on the copper electrodes.

### 3.3. X-ray Photoelectron Spectroscopy (XPS) Study

The perovskite-like crystalline structure of WO_3_ allows for withstanding a considerable amount of oxygen vacancies. This fact could even lead to structural changes with the stabilization of nonstoichiometric compounds, WO${}_{x}$, and/or to changes in the oxidation state of cations, for the sake of the charge balance [24,25]. X-ray photoemission spectroscopy is a very suitable tool to assess the electronic and chemical properties at the surface, and, in particular, the oxidation state of chemical elements. To this end, we have carried out XPS measurements on the ESCAmicroscopy beamline at Elettra on WO_3_ microstructures deposited on the electrode. The XPS analyses were carried out qualitatively in order to understand the band structure from the oxidation state of the sample for a later study of its optical properties.

Figure 6a shows the W 4f core level spectrum, which have contributions of both W${}^{6+}$ and W${}^{5+}$ oxidation states to the W 4f${}_{7/2}$ and the W 4f${}_{5/2}$ lines. The W${}^{6+}$ to W${}^{5+}$ ratio was deduced from the relative integrated areas of the corresponding peaks in the XPS spectra shown in Figure 6a, following the procedures described elsewhere [26] and yielding a 0.32 ratio. This could be evidence of the presence of defects, as oxygen vacancies close to the surface, or of some nonstoichiometric phases. The latter would not match with the above structural results that supports the γ, δ, and ϵ phases of WO_3_. However, the evident presence of W${}^{5+}$ would support a high concentration of oxygen vacancies at the surface level. On the other hand, the fitting of O 1s’ core level (Figure 6b) yields three peaks: 530.7, 531.5 and 532.8 eV peaks. The 530.7 eV peak is the most intense and is attributed to the O${}^{2-}$ ions of the WO_3_. The components at higher binding energies can be attributed to OH${}^{-}$ (531.5 eV) and O${}^{-}$ ions (532.8 eV) [27]. The hydroxide component is weak, but needed for an accurate fitting calculation, and could be originated from some water adsorption from the ambience. The component at the higher energy binding is associated with the presence of oxygen vacancies [27].

Therefore, the XPS results support that WO_3_ nanostructures contain a significant amount of oxygen vacancies, while keeping the structural arrangements of stoichiometric compounds. These defects often introduce electronic levels that behave as donors but also act as recombination sites that could hamper the performance of applications based on electron transfer, as photoelectrochemical water splitting or photocatalytic applications [28]. The donor electronic levels related to oxygen vacancies in oxides are usually involved in the luminescence mechanisms of the material. In highly defective samples, the associated band could even be the dominant emission. Herein, we have carried out cathodoluminescence measurements in the SEM, in order to excite all possible recombination paths by electron bombardment. The CL spectrum of WO_3_ nanostructures at room temperature reveals a broad emission in the visible range (1.6–2.9 eV) that could be deconvoluted into several components (Figure 7). Taking into account that the bandgap energy of bulk γ-WO_3_ has been reported to be about 2.62 eV, the complex CL spectrum achieved may have contributions from a number of electronic levels related to defects in the polymorphic WO_3_. Further work will be needed to more precisely understand the luminescence behaviour of WO_3_.

## 4. Conclusions

In this study, the polymorphism of micro-and nanostructures of WO_3_ produced by the resistive heating of metallic tungsten wires is demonstrated. The electrical transport through the metal wire was simulated taking into account the electrical and thermal transport through the wire, which allowed us to calculate the temperature profile along the wire. As a result, the density current conditions to promote the nucleation of WO_3_ micro- and nanostructures as a consequence of the atoms’ diffusion via the electromigration process were found. The structural and chemical characterization of the produced WO_3_ revealed that the material consists of a mixture of the γ (monoclinic I) phase, δ (triclinic) phase and ϵ (monoclinic II) phases at room temperature. The latter phases were reported at low temperatures, and only preserved up to room temperatures without a phase transformation under stress, mechanical treatments, ion implantation or nanostructuring processes. In addition, the WO_3_ nanostructures present a relevant amount of oxygen vacancies, which could affect the electronic band structures as the CL results suggest, and hence, their opto-electronic properties. Finally, the reported method of the resistive heating of metallic wires is shown as a promising route to produce oxide micro- and nanomaterials in a fast, easy and low-cost way. Additionally, it is a more sustainable type of synthesis with scaling-up potential.

## Figures and Tables

**Figure 1 nanomaterials-13-00884-f001:**
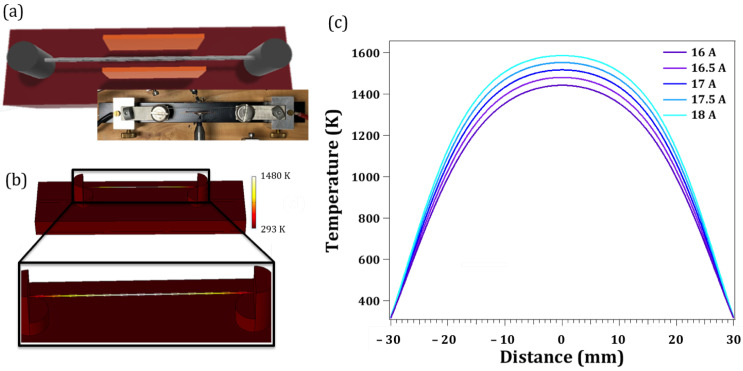
(**a**) Sketch of the setup (top view) for the resistive heating experiments. (**b**) Results of the temperature of the system simulated by COMSOL for 16.5 A current through the wire. (**c**) Temperature profiles along the W wire for several electrical currents.

**Figure 2 nanomaterials-13-00884-f002:**
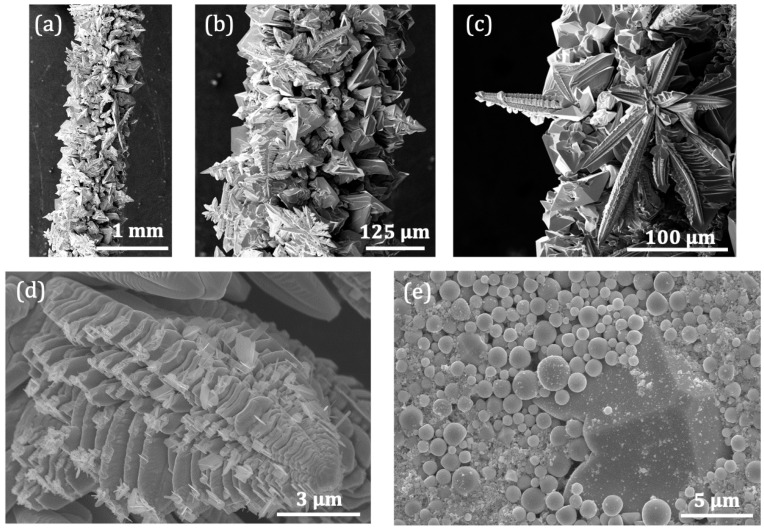
Secondary electron (SE) images of (**a**) WO_3_ microstructures formed on the W wire. (**b**) and (**c**) Zoom of the (**a**) SE image. (**d**) Detail of the micro- and nanostructures formed on the W wire. (**e**) SE image of nanostructures deposited on external copper electrodes located 3 mm apart from the wire.

**Figure 3 nanomaterials-13-00884-f003:**
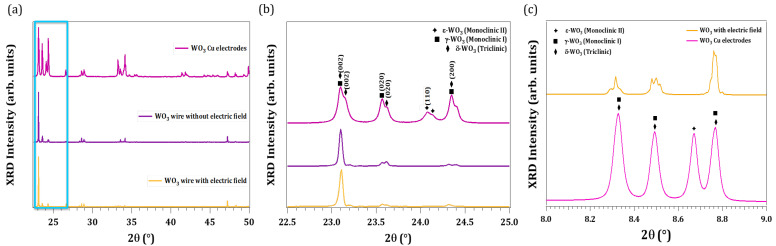
XRD results. (**a**) XRD data from three representative samples. Formed on W wire with electric field application (yellow) without field (purple), and on the Cu electrodes (pink). (**b**) Zoom XRD data with peaks assigned to ϵ, γ and δ-WO_3_ phases. (**c**) Synchrotron radiation XRD.

**Figure 4 nanomaterials-13-00884-f004:**
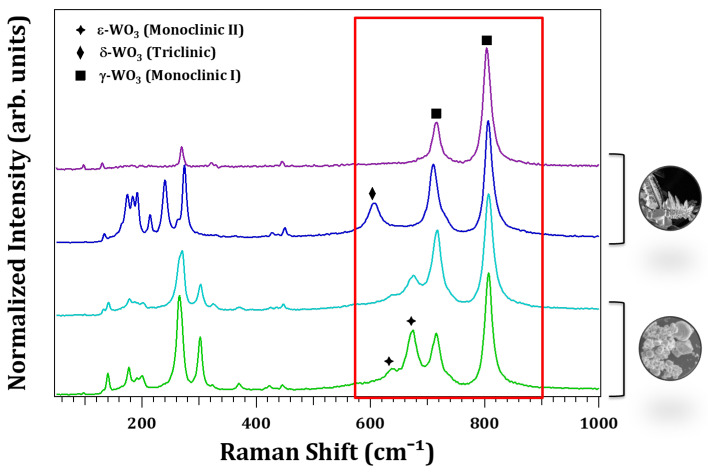
Comparison between Raman spectra from four representative samples in different structural states obtained on the surface of the W wire (purple and dark blue spectra) and on the Cu electrodes (green and blue spectra).

**Figure 5 nanomaterials-13-00884-f005:**
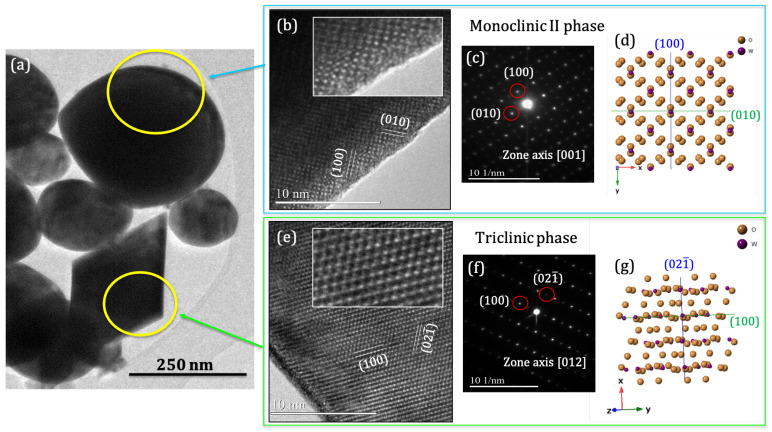
(**a**) TEM image of the obtained nanostructures on the copper electrodes. (**b**,**e**) HRTEM images. (**c**,**f**) SAED patterns with labelled diffracting planes according to [001] and [012] axis of the ϵ- and δ-phases, respectively. (**d**,**g**) Atomic arrangement of both crystalline structures according to the SAED data.

**Figure 6 nanomaterials-13-00884-f006:**
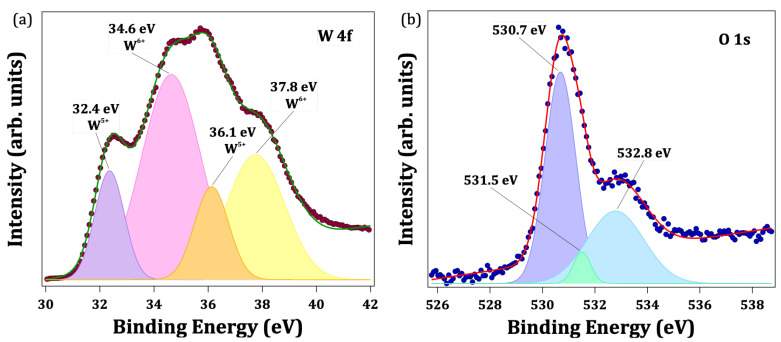
Fitted XPS core levels spectra of (**a**) W 4f and (**b**) O 1s.

**Figure 7 nanomaterials-13-00884-f007:**
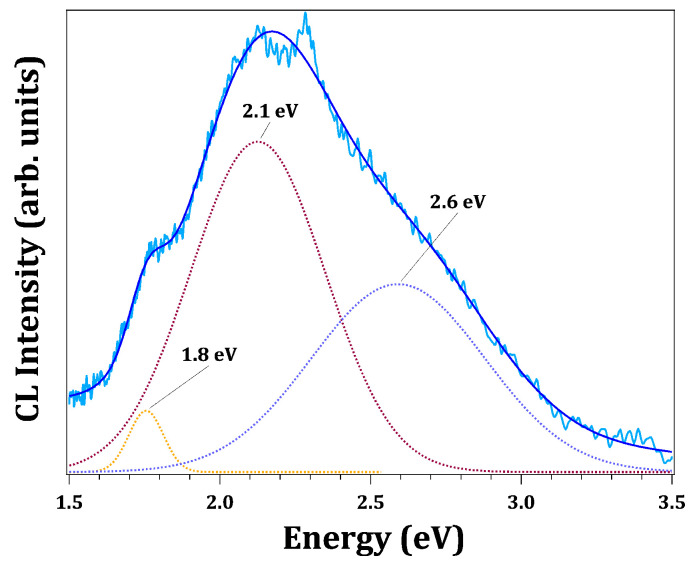
CL spectrum (light blue line) of WO_3_ nanostructures deposited on the copper electrodes. Data were deconvoluted into three Gaussian components (dotted lines). Fitted data (dark blue line).

## Data Availability

The data that support the findings of this study are available from the corresponding author upon reasonable request.

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
