# Peer review of "Room Temperature Polymorphism in WO3 Produced by Resistive Heating of W Wires"

_nanomaterials, 2023, doi:10.3390/nano13050884_

Round 1

Reviewer 1 Report

In this paper, the authors applied the Joule heating method to generate low-temperature phase WO3 on the W wire within seconds. WO3 with different phases could be deposited onto additional electrodes with an external electric field applied. The composition was analyzed in detail using XPS, XRD, SEM, TEM and other characterization methods. Overall, this is an interesting paper although there are some missing parts, making the story a bit unclear and difficult to repeat. I suggest publishing it after major revision.

Critical questions:

Lines 118-123, Page 4, the authors claimed that “The scale bar shows a local temperature of 1480 K at the centre of the wire. We have performed simulations at several currents to assess the local heating response of the W wire. Figure 1c shows the temperature profiles along the wire when density currents vary between 5 - 8 x 104A/cm2. It is observed that the central region of the wire, between 2 and 4 cm depending on the current, is well above 1000 K”. And a temperature distribution was also shown in the COMSOL simulation results in Figure 1b-c. Is the WO3 formation different at the edge of the W wire compared to the center due to the temperature difference?

Lines 196-198, Page 6, the authors claimed that “The resistive heating of W wires under suitable density currents and assisted by an external electric field, gives rise to a high amount of WO3 microstructures adopting several shapes.” Is it possible to clarify what currents are suitable for the growth? Besides this, does the annealing time play a role in the growth? Is the WO3 thickness related to the annealing temperature and time?

Figure 6, Page 7, Is it possible to list the percentage of different W- and O-containing species according to the XPS spectra? This could help make the results more clear.

Minor mistakes:

Line 53, page 2: “(all of them well below 3000 K)”. The “well” should be “were”.

In the methods part, detailed information of the instruments like the X-ray source of XRD and the laser wavelength of Raman should have been provided.

Reviewer 2 Report

The Auhtors reported fabrication of polymorphous WO3 via controlled Joule heating of tungsten wires.

The topic of the manuscript is within the scope of journal. The obtained results are interesting. 

However, several issues should be solved before the manuscript can be publishable in Nanomaterials.

- Was tungsten oxide obtained by using Joule heating o metallic tungsten before? If so, the appropriate citations should be provided. If not, the novelty aspect of the work should be provided.

- Figure 2. The morphological features of the obtained layers should be discussed. 

- What is the thickness of WO3 layer on W wire and Cu plates after the process? 

- If I understand correctly, the oxide structures shown in Fig. 2 were obtained by passing the current of 16.5 A for 50 s. Am I right? If so, why these conditions were applied? what about other conditions? Any effect of current and process duration is expected?

- Figure 3 - the labels for other signals (at 2theta > 25) should be also provided

- Figure 4 - the maxima attirbuted to γ-WO3 should be also indexed in the Raman spectra like other forms of WO3.

- Please consider transfer the only figure from Supplementary Information to the main manuscript.

- What is the main advantage of the proposed method compared to some other approaches (for instance, electrochemical ones?)

Round 2

Reviewer 1 Report

In this paper, the authors applied the Joule heating method to generate low-temperature phase WO3 on the W wire within seconds. WO3 with different phases could be deposited onto additional electrodes with an external electric field applied. The composition was analyzed in detail using XPS, XRD, SEM, TEM and other characterization methods. Overall, this is an interesting paper and the authors have addressed my questions. I suggest publishing this paper as it is now. 

Reviewer 2 Report

The authors adressed all issues raised by the reviewer, so now the manuscript is publishable.